# Three-Airy Beams, Their Propagation in the Fresnel Zone, the Autofocusing Plane Location, as Well as Generalizing Beams

Eugeny G. Abramochkin [1],[*], Svetlana N. Khonina [2],[3] and Roman V. Skidanov [2],[3]

1   Lebedev Physical Institute, Samara 443011, Russia
2   Image Processing Systems Institute, NRC "Kurchatov Institute", 151 Molodogvardeyskaya St., Samara 443001, Russia; khonina@ipsiras.ru (S.N.K.); romans@ipsiras.ru (R.V.S.)
3   Samara National Research University, 34 Moskovskoe Shosse, Samara 443086, Russia
*   Correspondence: ega@fian.smr.ru

**Abstract:** A family of 2D light fields consisting of the product of three Airy functions with linear arguments has been studied theoretically and experimentally. These fields, called three-Airy beams, feature a parameter shift and have a cubic phase and a super-Gaussian circular intensity in the far zone. Transformations of three-Airy beams in the Fresnel zone have been studied using theoretical, numerical, and experimental means. It has been shown that the autofocusing plane of a three-Airy beam is similar to the square root of the shift parameter. We also introduce generalized three-Airy beams containing nine free parameters, and obtain their Fourier transform in a closed form.

**Keywords:** Airy function; three-Airy beams; Fourier transform; Fresnel transform; autofocusing plane

## 1. Introduction

Light fields based on the Airy function usage are intensively studied objects in modern optics. As a solution of the 1D paraxial equation, the Airy function first appeared in 1979, when the non-diffracting Airy beams of infinite energy were found [1]. However, a boom in optical research related to the Airy function began in 2007, when it was noted that a linear exponential factor allows the construction of paraxial Airy beams of finite energy [2]. In general, Airy beams depend on some real parameters and are functionally stable solutions of the paraxial equation. Despite the lack of structural stability, under certain restrictions, there is a propagation zone in which the intensity of Airy beams varies very weakly. In addition, the parabolic shape of the propagation trajectory of Airy beams provides new opportunities for the practical application of such beams since traditional Gaussian beams are characterized by rectilinear propagation.

This discovery was followed by studies and publications that looked at various aspects related to Airy beams. Among the theoretical works, the following results should be noted: the transformation of Airy–Gaussian beams in first-order optical systems [3], the Poynting vector and orbital angular momentum of Airy beams [4], the nonparaxial study of Airy beams [5,6], truncated Airy beams [7,8], the comparison of Airy beams and Hermite–Gaussian modes under astigmatic transformation [7], vortex Airy beams [7,9] and others [10–14]. Among the experimental works, we mention papers related to the study of the propagation of Airy beams in various media [15–17], as well as the use of Airy beams for problems related to microparticles [18] (see also the references therein).

As a rule, either 1D Airy beams or their 2D analogs, obtained by multiplying two 1D beams whose arguments are orthogonal Cartesian coordinates in the plane, were considered. Airy beams of more complex shape (for example, circular Airy beams) have been investigated only numerically and experimentally since theoretical study requires the discovery of quite nontrivial integrals in a closed form. However, the theory of integral transformations of Airy functions is not yet sufficiently developed to cope with such challenges.

One of the nontrivial examples when theoretical research turned out to be possible is a family of three-Airy beams [19]. In 2010, these beams were introduced as the product of three 1D Airy patterns, which are shifted from the origin to the vertices of an equilateral triangle and rotated so that the resulting field is invariant under rotation by $120°$:

$$\text{Ai}_3(\mathbf{r}\,|\,c) = \text{Ai}\left(\frac{-x\sqrt{3}-y}{2}+c\right)\text{Ai}\left(\frac{x\sqrt{3}-y}{2}+c\right)\text{Ai}(y+c), \qquad (1)$$

where

$$\text{Ai}(x) = \frac{1}{2\pi}\int_{\mathbb{R}}\exp\left(\frac{\text{i}t^3}{3}+\text{i}xt\right)\text{d}t \qquad (2)$$

is the Airy function definition, $\mathbf{r} = (x, y)$ is a 2D vector, and $c$ is a shift parameter.

The shift parameter determines the shape of the three-Airy beams and their behavior when propagating in the Fresnel zone. As it turned out, the Fourier transform of these beams has a cubic phase and a circular super-Gaussian intensity [19]:

$$\mathcal{F}\left[\text{Ai}_3(b\boldsymbol{\rho}\,|\,c)\right](\mathbf{r}) = \frac{1}{3^{5/6}\pi b^2}\exp\left(-\frac{2\text{i}(3x^2y-y^3)}{27b^3}\right)\text{Ai}\left(3^{2/3}c + \frac{2|\mathbf{r}|^2}{3^{4/3}b^2}\right). \qquad (3)$$

Here, $\boldsymbol{\rho} = (\xi, \eta)$ is a 2D vector, $b$ is a scaling factor,

$$\mathcal{F}\left[f(\boldsymbol{\rho})\right](\mathbf{r}) = \frac{1}{2\pi}\iint_{\mathbb{R}^2}\text{e}^{-\text{i}\langle\mathbf{r},\boldsymbol{\rho}\rangle}f(\boldsymbol{\rho})\,\text{d}^2\boldsymbol{\rho} \qquad (4)$$

is 2D Fourier transform, and $\langle\mathbf{r},\boldsymbol{\rho}\rangle = x\xi + y\eta$ is the standard scalar product of 2D vectors. As is seen in (3), only the Airy factor depends on the shift parameter. A simple proof of the relation (3) is presented in Appendix A.

The most interesting cases are when, at the central point, $x = y = 0$, the Airy function reaches a maximum (i.e., $3^{2/3}c = a'_n$) or goes to zero (i.e., $3^{2/3}c = a_n$). Here $a_n$ and $a'_n$ are zeros of the functions $\text{Ai}(x)$ and $\text{Ai}'(x)$, respectively. All of them are real and located on the negative semi-axis. Asymptotic expansions are known for both families of zeros [20],

$$a_n \approx -\left(\frac{3\pi}{8}(4n-1)\right)^{2/3}, \quad a'_n \approx -\left(\frac{3\pi}{8}(4n-3)\right)^{2/3}. \qquad (5)$$

Even for small $n$, the initial terms of these expansions give a very good approximation to the exact values of the zeros. For example,

$$\begin{aligned}
a_1 &= -2.338\ldots \approx -2.320, & a'_1 &= -1.018\ldots \approx -1.115, \\
a_2 &= -4.087\ldots \approx -4.081, & a'_2 &= -3.248\ldots \approx -3.261, \\
a_3 &= -5.520\ldots \approx -5.517, & a'_3 &= -4.820\ldots \approx -4.826, \\
a_4 &= -6.786\ldots \approx -6.784, & a'_4 &= -6.163\ldots \approx -6.167,
\end{aligned} \qquad (6)$$

where the first number is the exact value of zero, and the second is due to Equation (5).

In subsequent years, three-Airy beams were constructed and studied in various ways. Torre in [21] found their intensity moments of second order. In [22], the three-Airy beam for the case $3^{2/3}c = a'_1$ was implemented on the basis of the Fourier transform method. Its propagation in free space and a nonlinear medium (SBN crystal) was studied experimentally. In [23], Desyatnikov et al. also used the Fourier transform method, but the initial field was formed by the holographic method. In that article, two beams with parameters $3^{2/3}c = a_3$ and $3^{2/3}c = a'_3$ were investigated, and autofocusing of three-Airy beams was discussed. In addition, Desyatnikov considered the propelling behavior of such beams when an optical singularity is embedded, as well as the prospects for their use for transporting particles. In [24], vortex three-Airy beams with different topological charges have been investigated.

In this work, we continue to study three-Airy beams by theoretical and numerical means, as well as in optical experiments. In Section 2, we discuss the ways to evaluate the Fresnel transform of these beams. Here, the experimental implementation of three-Airy beams is also considered. In Section 3, we investigate the autofocusing behavior of these beams. In Section 4, we generalize the relation (3) and provide some corollaries concerning the beams of finite and infinite energy. In Conclusion, we discuss possible applications of the results obtained.

We consider the study of three-Airy beams at various values of the shift parameter, the characteristics of their propagation in the Fresnel zone, and the determination of the autofocusing plane depending on the shift parameter as a necessary basis for the subsequent transition to the study of multiple beams of Pearcey, swallowtail, and other diffraction catastrophes, whose non-Gaussian nature also leads to the appearance of an autofocusing plane when propagating in the Fresnel zone.

## 2. Three-Airy Beams in the Fresnel Diffraction Zone

### 2.1. Theory

If a coherent light field of a wavelength $\lambda$ is specified by its complex amplitude $F_0(\mathbf{r})$ in the initial plane $z = 0$, then the field propagation in free space is described in the paraxial approximation by the equation $\{\partial_x^2 + \partial_y^2 + 2ik\partial_z\}F(\mathbf{r}, z) = 0$, where $F(\mathbf{r}, 0) = F_0(\mathbf{r})$. If $F_0(\mathbf{r})$ vanishes sufficiently rapidly as $|\mathbf{r}| \to \infty$, then $F(\mathbf{r}, z)$ related to $F_0(\mathbf{r})$ by the Fresnel transform [25]:

$$F(\mathbf{r}, z) = \mathbf{FR}_z[F_0(\boldsymbol{\rho})](\mathbf{r}) = \frac{k}{2\pi i z} \iint_{\mathbb{R}^2} \exp\left(\frac{ik}{2z}|\mathbf{r} - \boldsymbol{\rho}|^2\right) F_0(\boldsymbol{\rho}) \, d^2\boldsymbol{\rho}. \tag{7}$$

where $k = 2\pi/\lambda$ is the wavenumber of light.

The numerical evaluation of the double integral (7) can be carried out by expanding the initial field into a series of Hermite–Gaussian (HG) modes (see Appendix B).

When choosing $F_0(\boldsymbol{\rho}) = \mathrm{Ai}_3(b\boldsymbol{\rho} \,|\, c)$, the question arises: what Gaussian beam width $w$ is optimal for approximating a three-Airy beam by the superposition of HG modes (A11) if the double series is replaced by a truncated finite sum?

$$\mathbf{funcFR}_{z,w}\big[\mathrm{Ai}_3(b\boldsymbol{\rho} \,|\, c)\big](\mathbf{r}) \cong \sum_{\substack{0 \leq n \leq N \\ 0 \leq m \leq M}} c_{n,m} e^{-i(n+m)\arg\sigma} \mathcal{H}_{n,m}\left(\frac{\mathbf{r}}{w}\right). \tag{8}$$

This problem can be solved numerically using the least squares method (see [19] for detail), selecting the shift parameter $c = 3^{-2/3}a_1'$, for which the Fourier image of the three-Airy beam has a circular intensity very similar to Gaussian. The solution is $w = 1.027/b$, and this value remains optimal even if $3^{2/3}c \neq a_1'$.

Figure 1 depicts the intensity and phase distributions of the beam (8) for the shift parameters $c = 3^{-2/3}a_3'$ and $c = 3^{-2/3}a_3$. Other parameters are $b = 1$, $w = 1$ and $z \in [0, \pi/2]$. As can be seen, when the three-Airy beam propagates, there is a plane in which the intensity has a trefoil shape with a strong central peak. We consider this plane to be the autofocusing (AF) plane.

For the initial beam $F_0(\boldsymbol{\rho}) = \mathrm{Ai}_3(b\boldsymbol{\rho} \,|\, c)$, we were unable to calculate analytically the double integral (7), while we reduce it to a 1D integral:

$$\mathbf{FR}_z\big[\mathrm{Ai}_3(b\boldsymbol{\rho} \,|\, c)\big](\mathbf{r})$$

$$= \frac{1}{\sqrt{\pi}} \exp\left(\frac{9icb^2z}{4k} + \frac{27ib^6z^3}{64k^3}\right) \int_{\mathbb{R}} e^{-t^2} \mathrm{Ai}_3\left(b\mathbf{r} \,\Big|\, c + \frac{9b^4z^2}{16k^2} + \sqrt{\frac{b^2z}{k}} \cdot e^{-\pi i/4}t\right) dt. \tag{9}$$

This integral representation is useful in the theoretical study of the elliptic umbilic catastrophe [26], but it is not applicable for numerical simulation related to the propagation of three-Airy beams in the Fresnel zone. The reason is that the integrand in (9) is the product

of a rapidly decreasing Gaussian function, $e^{-t^2}$, and a rapidly increasing Airy function of the complex argument, $Ai_3 \sim e^{|t|^{3/2}}$ as $t \to -\infty$. Of course, the integral (9) converges absolutely. However, to obtain acceptable calculation accuracy, we are forced to make the integration domain too large, which leads to an excessive increase in calculation time. As a result, for numerical simulation, the use of the series expansion in terms of HG modes (8) is preferable.

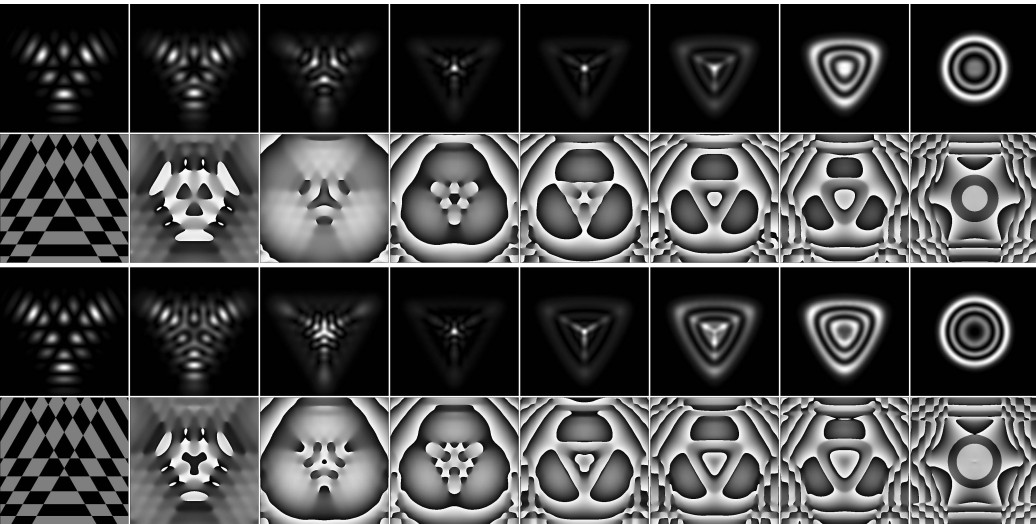

**Figure 1.** Numerically evaluated intensity and phase distributions of the beam (8), where $c = 3^{-2/3}a_3'$ (top two rows) and $c = 3^{-2/3}a_3$ (bottom two rows). The leftmost frames correspond to the initial plane $z = 0$, while the rightmost ones correspond to the Fourier plane $z = \infty$. All frames are shown in the square $[-h, h] \times [-h, h]$, where $h = 4.0$ for the top row and $h = 4.5$ for the bottom. As usual, the $x$ axis is horizontal, the $y$ axis is vertical.

### 2.2. Experiment

The experimental part of our investigation of three-Airy beams has been conducted as follows. Figure 2 shows the optical set-up. We use solid-state laser $L$ with a wavelength of 532 nm. Lenses $L_1$ and $L_2$ ensure beam expansion to the desired size. Lens $L_3$ forms intensity distribution on the sensitive area of the video camera. To reduce the formation distance, a lens with a focal length of 1000 mm was used.

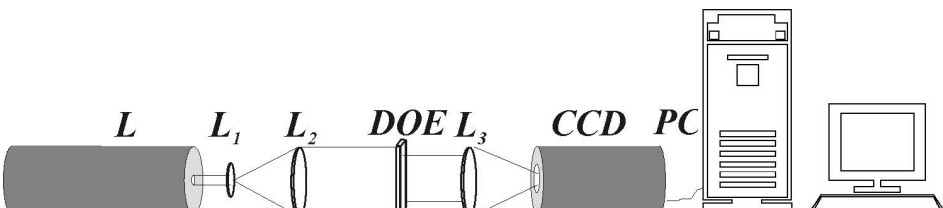

**Figure 2.** Experimental set-up: $L$ is laser, $L_1$ is microlens 20×, $L_2$ is lens with $f = 400$ mm, $L_3$ is lens with $f = 1000$ mm, *CCD* is video camera VSTT-252.

The amplitude–phase distribution presented in Figure 3a,b was encoded into a pure phase one by partial encoding method [27]. The partial encoding technique makes it possible to vary the error of the field generation and the diffraction efficiency in a wide range, choosing the best ratio for a specific task. A threshold value $\alpha \in [0, 1]$ is proportional to the accuracy of the formation of a given field and inversely proportional to efficiency. The simulation showed a slight difference in the formed patterns calculated for $\alpha = 1$ and $\alpha = 0.5$, so in the experiments, we used an optical element for $\alpha = 0.5$ (see Figure 3d), because in this case, higher diffraction efficiency is ensured.

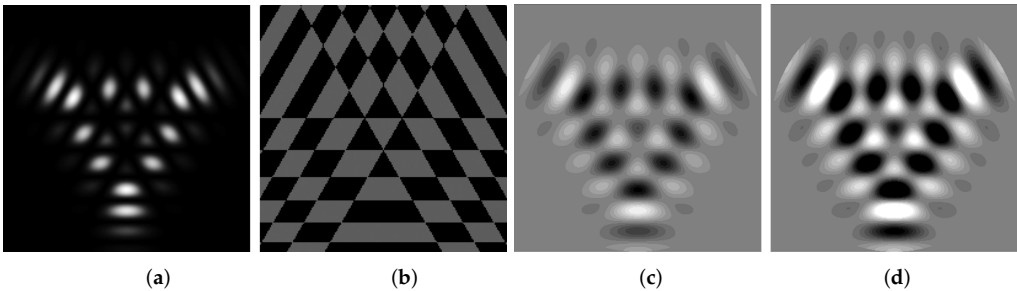

| (**a**) | (**b**) | (**c**) | (**d**) |

**Figure 3.** Numerically evaluated intensity (**a**) and phase (**b**) of the input three-Airy beam. Phase of an optical element with encoded amplitude for the formation of a three-Airy beam at $\alpha = 1$ (**c**) and $\alpha = 0.5$ (**d**).

The DOE shown in Figure 3d was fabricated using the method photolithography on a quartz plate with a resolution of 1 μm, and the microrelief depth was 530 nm. Figure 4 shows the shape of the surface microrelief section DOE. The diameter of the DOE is 4 mm.

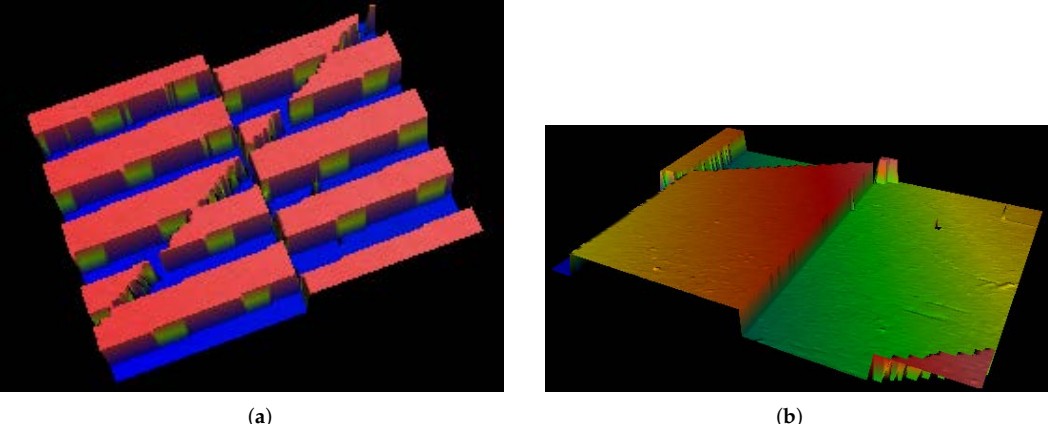

| (**a**) | (**b**) |

**Figure 4.** Surface shape of the coded section of the DOE microrelief (**a**), surface shape of the uncoded section of the DOE microrelief (**b**).

The three-Airy beam propagation simulation and corresponding experimentally recorded intensity patterns obtained for phase element (Figure 3d), illuminated by a converging beam (with a focus at a distance of 1000 mm from the input plane), are shown in Figure 5.

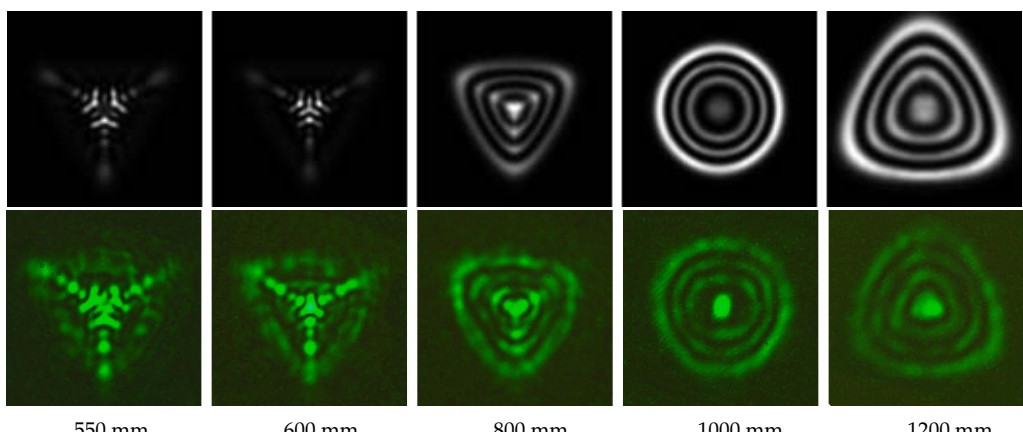

| 550 mm | 600 mm | 800 mm | 1000 mm | 1200 mm |

**Figure 5.** Intensity distribution of three-Airy beam for the shift parameter $c = 3^{-2/3}a'_4$ at various distances from the input plane: numerical simulations (**top row**) and results of optical experiment (**bottom row**).

### 3. Autofocusing Plane of Three-Airy Beams

During free-space propagation, many non-Gaussian paraxial light fields demonstrate an autofocusing (AF) behavior. It means that a light beam propagating in the Fresnel zone and with a wide intensity profile at some point suddenly changes sharply and takes the form of a single, highly localized peak of large magnitude.

The most famous beams with the AF property are circular Airy beams $e^{a(r_0-r)}\text{Ai}(r_0 - r)$, where $r_0$ is the radius of the Airy ring on the input plane and $a$ is an exponential truncation factor [28,29] (see also [30] and the references therein). Due to the radially symmetrical shape of such fields, the study of the location of their AF plane when propagating in the Fresnel zone is reduced to a 1D problem and allows the application of asymptotical methods based on the idea of the stationary phase.

For three-Airy beams, $\text{Ai}_3(\mathbf{r}\,|\,c)$, even the very definition of the concept of "autofocusing plane" becomes a problem since the transverse intensity distribution of such beams is not radially symmetric and, as a consequence, a 3D profile of its intensity is not bell-shaped. However, in the simplest version, when the shift parameter is defined by the relation $3^{2/3}c = a_1' \approx -1.018$, the intensity of the three-Airy beam in the Fresnel zone changes weakly, transforming from triangular to circular shape. In this case, the intensity reaches a maximum value on the axis of the beam propagation.

Thus, there are two ways to find the location of the AF plane of the three-Airy beam. First, one can write an equation for its intensity extrema points located on the optical axis (i.e., at $\mathbf{r} = \mathbf{0}$):

$$\partial_z \left| \text{Ai}_3(\mathbf{0}, z\,|\,c) \right|^2 = 0 \tag{10}$$

and try to solve it within the framework of some additional asymptotic assumptions. Second, one can represent the Fresnel transform of the three-Airy beam as a certain integral of the pure phase exponential and apply the stationary phase method to find the caustic. We consider both of them.

#### 3.1. Three-Airy Beam Intensity at the Points of the Optical Axis

It is known that the Fresnel transform (7) of a field $F_0(\boldsymbol{\rho})$ can be written in terms of the Fourier transform of the field. Specifically, since

$$F_0(\boldsymbol{\rho}) = \mathcal{F}^{-1}\left[ \mathcal{F}[F_0](\mathbf{r}) \right](\boldsymbol{\rho}) = \frac{1}{2\pi} \iint_{\mathbb{R}^2} e^{i\langle \mathbf{r}, \boldsymbol{\rho}\rangle} \mathcal{F}[F_0](\mathbf{r})\, d^2\mathbf{r}, \tag{11}$$

then

$$\mathbf{FR}_z\left[ F_0(\boldsymbol{\rho}) \right](\mathbf{r}) = \mathcal{F}^{-1}\left[ \exp\left( -\frac{iz}{2k}|\boldsymbol{\rho}|^2 \right) \mathcal{F}[F_0](\boldsymbol{\rho}) \right](\mathbf{r}). \tag{12}$$

Substituting the three-Airy beam $F_0(\boldsymbol{\rho}) = \text{Ai}_3(b\boldsymbol{\rho}\,|\,c)$ and its Fourier image (3), one obtains

$$\text{Ai}_3(b\mathbf{r}, z\,|\,c) = \frac{1}{3^{5/6}\pi b^2} \mathcal{F}^{-1}\left[ \exp\left( -\frac{iz}{2k}|\boldsymbol{\rho}|^2 - \frac{2i(3\xi^2\eta - \eta^3)}{27b^3} \right) \text{Ai}\left( 3^{2/3}c + \frac{2|\boldsymbol{\rho}|^2}{3^{4/3}b^2} \right) \right](\mathbf{r}). \tag{13}$$

Let us simplify this integral for $\mathbf{r} = \mathbf{0}$, passing to polar coordinates, $\xi + i\eta = \rho e^{i\phi}$, integrating over the polar angle and introducing the variable $t = \rho^2/9b^2$:

$$\text{Ai}_3(\mathbf{0}, z\,|\,c) = \frac{3^{7/6}}{2\pi} \int_0^\infty \exp\left( -\frac{9izb^2}{2k}t \right) J_0(2t^{3/2}) \text{Ai}(3^{2/3}[c + 2t])\, dt. \tag{14}$$

Then the equation $\partial_z |\text{Ai}_3(\mathbf{0}, z\,|\,c)|^2 = 0$ takes the form

$$\text{Im}\{G_1^* \cdot G_0\} = 0, \quad \text{where} \quad G_m = \int_0^\infty \exp\left( \frac{9izb^2}{2k}t \right) J_0(2t^{3/2}) \text{Ai}(3^{2/3}[c + 2t]) t^m\, dt, \tag{15}$$

and the asterisk means complex conjugation.

Some intensity curves, $|\text{Ai}_3(\mathbf{0}, z \,|\, c)|^2$, are depicted in Figures 6 and 7. Here, we use the dimensionless variable $z := 2zb^2/k$ and select the shift parameter $c$ so that the Fourier image of the three-Airy beam reaches its maximum value at a point lying on the optical axis, $c = c_n = 3^{-2/3} a'_n$. Thus, the location of the AF plane $z_s(c_n)$ depends on $n$. A numerical study of the discrete function $z_s(c_n)$, where $20 \le n \le 150$, revealed that $z_s(c_n) \approx \sqrt{n} + 5$ for large $n$ (in particular, $z_s(c_{60}) - \sqrt{60} = 4.812$), while, of course, the numerical results cannot be considered to be a true asymptotic behavior of $z_s(c)$ as $c \to \infty$.

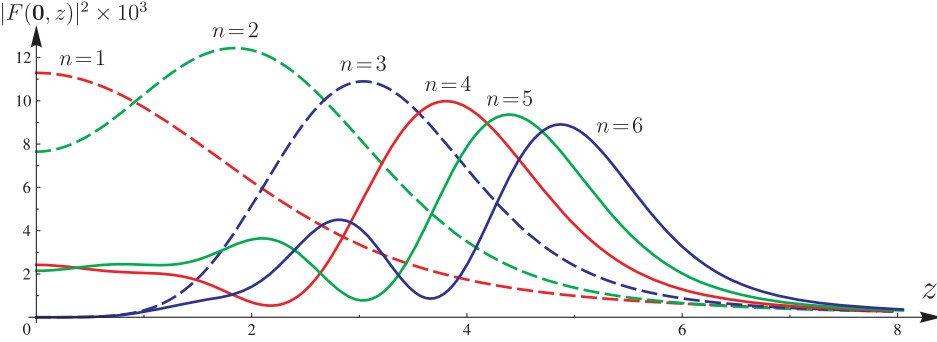

**Figure 6.** Intensity distributions $|\text{Ai}_3(\mathbf{0}, z \,|\, c)|^2$ versus dimensionless $z := 2zb^2/k$ when the shift parameter is $c = c_n = 3^{-2/3} a'_n$, $1 \le n \le 6$. The global maximum point characterizes the location of the AF plane $z_s(c_n)$. If $n = 1$, then there is no such plane. For other $n$s, the plane exists and the value of $z_s(c_n)$ increases monotonically with increasing $n$: 1.842, 3.038, 3.806, 4.389, and 4.868.

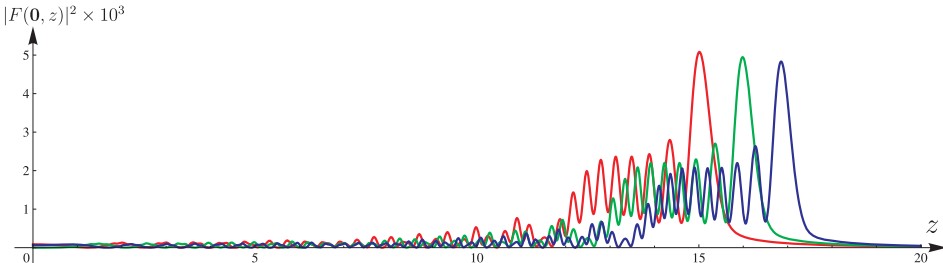

**Figure 7.** Intensity distributions $|\text{Ai}_3(\mathbf{0}, z \,|\, c)|^2$ versus dimensionless $z := 2zb^2/k$ when the shift parameter is $c = 3^{-2/3} a'_n$, $n = 100$ (red), 120 (green), and 140 (blue). Here, $z_s = 15.012$, 15.985, and 16.852, respectively.

### 3.2. Caustic of Three-Airy Beam upon Propagation in the Fresnel Zone

Finding the caustic is one of the traditional problems that arise when the properties of diffraction catastrophe integrals and light fields associated with them are studied. Each diffraction catastrophe is connected to a definite caustic due to the equations on the stationary phase and the Hessian. A good introduction to this field (catastrophe optics) is the paper [31] (see also [32] and the references therein). Recently, caustical investigation has been applied to aberration laser beams [33], vortex beams [34], and Ince–Gaussian beams [35].

Let us consider the problem of finding the AF plane of the beam $\text{Ai}_3(\mathbf{r}, z \,|\, c)$ as the process of the appearance of a caustic point of the beam when it propagates in the Fresnel zone. Replacing the radial Airy function in Equation (13) with the definition (2) and changing the variable $t \to 3^{-2/3} t$, we obtain

$$\text{Ai}_3(\mathbf{r}, z \,|\, c) = \frac{3^{-5/6}}{4\pi^3} \iiint_{\mathbb{R}^3} \exp\left(iP(\xi, \eta, t)\right) d\xi \, d\eta \, dt, \qquad (16)$$

where $P(\xi, \eta, t) = (x\xi + y\eta) - \frac{1}{4}z(\xi^2 + \eta^2) - \frac{2}{27}(3\xi^2\eta - \eta^3) + \frac{1}{27}t^3 + t\left[c + \frac{2}{9}(\xi^2 + \eta^2)\right]$.

By making the substitution of the output variables $(x, y, z, c) = \frac{4}{9}(X, Y, 2Z, -A)$ in the function $P(\xi, \eta, t)$,

$$P(\xi, \eta, t) = \tfrac{4}{9}(X\xi + Y\eta - At) + \tfrac{2}{9}(t - Z)(\xi^2 + \eta^2) + \tfrac{1}{27}(t^3 - 6\xi^2\eta + 2\eta^3), \quad (17)$$

and equating its gradient to zero, $\partial_\xi P = \partial_\eta P = \partial_t P = 0$, we obtain a system for finding the stationary points:

$$\xi\eta + (Z - t)\xi = X, \quad \tfrac{1}{2}(\xi^2 - \eta^2) + (Z - t)\eta = Y, \quad (\xi^2 + \eta^2) + \tfrac{1}{2}t^2 = 2A. \quad (18)$$

From the last equation, it is seen that a real solution $(\xi, \eta, t)$ exists for $A > 0$ only.

Equating the Hessian of $P(\xi, \eta, t)$ to zero, we have a fourth relationship between the variables:

$$(t^3 - 6\xi^2\eta + 2\eta^3) + 2Z(\xi^2 + \eta^2 - t^2) + t(Z^2 - 3[\xi^2 + \eta^2]) = 0. \quad (19)$$

This equation, when jointly solved with the system (18), allows us to find the location and shape of the caustic curve.

Ideally, our further plan of action is as follows. We find an explicit solution of the system (18) in the form of $\xi = \xi(X, Y, Z, A)$, $\eta = \eta(X, Y, Z, A)$, and $t = t(X, Y, Z, A)$. Then, we substitute it into (19) and obtain a caustic equation, $\Phi(X, Y, Z, A) = 0$, which for the values $X = Y = 0$ leads to AF plane: $\Phi(0, 0, Z, A) = 0 \Rightarrow Z = Z(A)$.

Carrying out this plan and passing from Cartesian coordinates to polar ones, $(\xi, \eta) = (\rho \cos \phi, \rho \sin \phi)$, we obtain

$$X = \tfrac{1}{2}\rho^2 \sin 2\phi + (Z - t)\rho \cos \phi, \quad Y = \tfrac{1}{2}\rho^2 \cos 2\phi + (Z - t)\rho \sin \phi, \quad \rho^2 + \tfrac{1}{2}t^2 = 2A,$$
$$(t^3 - 2\rho^3 \sin 3\phi) + 2Z(\rho^2 - t^2) + t(Z^2 - 3\rho^2) = 0.$$

Later, we use the equation $\rho^2 + \tfrac{1}{2}t^2 = 2A$ to replace $\rho = \sqrt{2A} \cos \theta$, $t = 2\sqrt{A} \sin \theta$, where $-\pi/2 \le \theta \le \pi/2$. In fact, it makes the transition from Cartesian to spherical coordinates $(\xi, \eta, t) \to (\rho, \phi, \theta)$.

Making another change in variables, $(X, Y, Z) = (A\tilde{X}, A\tilde{Y}, \sqrt{A}\,\tilde{Z})$, we obtain the caustic equations *without* the explicit presence of the variable $A$ (thus, the use of the stationary phase method at $A \gg 1$ becomes justified):

$$\tilde{X} = \cos^2 \theta \sin 2\phi + \sqrt{2}(\tilde{Z} - 2\sin\theta)\cos\theta \cos\phi, \quad (20)$$

$$\tilde{Y} = \cos^2 \theta \cos 2\phi + \sqrt{2}(\tilde{Z} - 2\sin\theta)\cos\theta \sin\phi, \quad (21)$$

$$2\sqrt{2}\cos^3 \theta \sin 3\phi = \sin\theta(\tilde{Z} - 2\sin\theta)^2 + 2\cos^2\theta(\tilde{Z} - 3\sin\theta). \quad (22)$$

From the last equation, we find the solution $\phi = \phi(\tilde{Z}, \theta)$. Substituting it into Equations (20) and (21), we obtain the caustic curve in parametric form.

The case $\cos \theta = 0$ is the most interesting. If $\theta = -\pi/2$, then there is no solution due to the constraint $z \ge 0$. If $\theta = \pi/2$, then we obtain the singular solution $(\tilde{X}, \tilde{Y}, \tilde{Z}) = (0, 0, 2)$. This is a caustic point. It determines the location of the AF plane. Returning to the original variables $(x, y, z)$, we obtain $x = y = 0$ and

$$z_s = \tfrac{8}{3}\sqrt{-c}. \quad (23)$$

Thus, if the shift parameter $c$, while remaining negative, increases in absolute value, then the AF plane moves away from the initial plane.

It is interesting to note that the same result is obtained if we replace the Bessel function in Equation (15) with its asymptotic expression, $J_0(x) \sim \sqrt{2/\pi x} \cos(x - \pi/4)$. However, this way requires much more effort.

In the numerical experiments above, we considered the case $c = c_n = 3^{-2/3}a'_n$. Then,

$$z_s(c_n) = \tfrac{8}{3}\sqrt{-3^{-2/3}a'_n} \overset{(5)}{\approx} \tfrac{4}{3}\big[\pi(4n - 3)\big]^{1/3}. \tag{24}$$

The exponent of $n$ in the last formula does not coincide with the analogous indicator in the dependence $z_s(c_n) \approx \sqrt{n} + 5$, found empirically for $n \leq 150$ in numerical experiments. Nevertheless, already at $n = 100$, we obtain a quite good correspondence with an exact numerical result (see Figure 6): $z_s(c_{100}) \overset{(24)}{=} 14.3522 \approx 15.012$. Taking into account the *asymptotic* nature of the formula (23), we expect that its accuracy will increase with increasing $n$, while the accuracy of the empirical formula will decrease.

Fixing $\tilde{Z} = 2$ (the AF plane), let us investigate the question: what is the shape of the caustic curve in the plane of variables $(\tilde{X}, \tilde{Y})$? We omit the case $\cos\theta = 0$, leading to the caustic point. Then,

$$\sin 3\phi = \frac{\sin\theta(2 - 2\sin\theta)^2 + 2\cos^2\theta(2 - 3\sin\theta)}{2\sqrt{2}\cos^3\theta}. \tag{25}$$

The curve of the RHS of (25) versus $\theta \in [-\pi/2, \pi/2]$ is shown in Figure 8. As is seen, there are two intervals, $\theta \in [-0.651, -0.418]$ and $[0.339, 1.123]$, for which the RHS value is placed in $[-1, 1]$, i.e., for which a solution $\phi_0 = \phi(\theta)$ exists. The general solution $\phi(\theta)$ is a set of the values $\{\phi_0, \phi_0 + \tfrac{2\pi}{3}, \phi_0 + \tfrac{4\pi}{3}\}$ and $\{\tfrac{\pi}{3} - \phi_0, \pi - \phi_0, \tfrac{5\pi}{3} - \phi_0\}$. If we use these six values $\phi(\theta)$ to construct the parametric curve given by Equations (20) and (21), then we obtain a caustic, consisting of two parts:

- outside—a triangular-shaped curve (six values for $\theta \in [-0.651, -0.418]$ are used),
- inside—a 3-bladed curve with sharp vertices (six values for $\theta \in [0.339, 1.123]$ are used).

The shape of the caustic (shown on the right in Figure 8) is very similar to what is obtained in numerical and optical experiments.

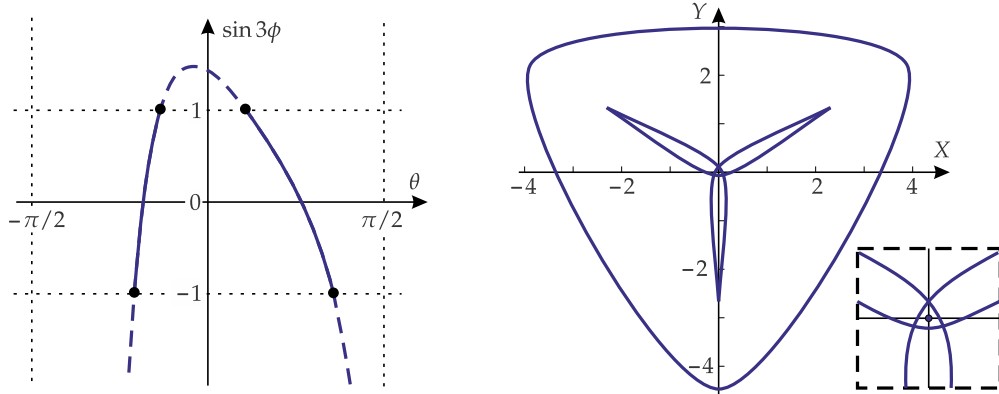

**Figure 8.** Left: intervals in $\theta$ for which there is a solution $\phi(\theta)$ to Equation (25). Right: caustic $\tilde{X} = \tilde{X}(\phi, \theta(\phi))$, $\tilde{Y} = \tilde{Y}(\phi, \theta(\phi))$, $0 \leq \phi \leq 2\pi$, obtaining from Equations (20)–(22) with $\tilde{Z} = 2$. Callout: a zoomed-in area around the origin shows the caustic point, which exists only in the AF plane.

## 4. Fourier Transform of Generalized Three-Airy Beams

In this section, we consider a 2D light field generalizing the three-Airy beam (1). Let us define the field as the product of three Airy functions with linear arguments of general form,

$$\mathrm{Ai}_3(\mathbf{r}\,|\,\mathbf{M}) = \prod_{1 \leq n \leq 3} \mathrm{Ai}(A_n x + B_n y + C_n), \tag{26}$$

where $A_n$, $B_n$, $C_n$ are real parameters combined into a matrix

$$\mathbf{M} = \begin{pmatrix} A_1 & A_2 & A_3 \\ B_1 & B_2 & B_3 \\ C_1 & C_2 & C_3 \end{pmatrix}. \tag{27}$$

We do not discuss what values these parameters can take. The only restriction is that each of the Airy multipliers is nontrivial (i.e., we discard cases like $A_1 = B_1 = 0$).

The main result of this section is that the Fourier image of the field (26) can be found in a closed form:

$$\mathcal{F}\big[\text{Ai}_3(\boldsymbol{\rho} \,|\, \mathbf{M})\big](\mathbf{r}) = \frac{1}{2\pi|\Delta|^{1/3}} \exp(iP_1 + iP_3)\text{Ai}(P_0 + P_2), \tag{28}$$

where $\Delta$ is a constant and $P_n = P_n(\mathbf{r})$ are homogeneous real-valued polynomials of degree $n$. All of them are quite cumbersome since the LHS of (28) contains 9 free parameters. We write $\Delta$ and $P_n$ using auxiliary quantities. Specifically, let

$$\Delta_1 = \begin{vmatrix} A_2 & A_3 \\ B_2 & B_3 \end{vmatrix}, \quad \Delta_2 = \begin{vmatrix} A_3 & A_1 \\ B_3 & B_1 \end{vmatrix}, \quad \Delta_3 = \begin{vmatrix} A_1 & A_2 \\ B_1 & B_2 \end{vmatrix}, \tag{29}$$

that means that $\det \mathbf{M} = C_1\Delta_1 + C_2\Delta_2 + C_3\Delta_3$. Let also $R_n = B_n x - A_n y$. Then,

$$\Delta = \Delta_1^3 + \Delta_2^3 + \Delta_3^3, \quad P_0 = \frac{\det \mathbf{M}}{\Delta^{1/3}}, \quad P_2 = \frac{\Delta_1\Delta_2\Delta_3}{\Delta^{4/3}} \sum_{1 \leq n \leq 3} \frac{R_n^2}{\Delta_n},$$

$$P_1 = \frac{1}{\Delta} \begin{vmatrix} R_1 & R_2 & R_3 \\ \Delta_1^2 & \Delta_2^2 & \Delta_3^2 \\ C_1 & C_2 & C_3 \end{vmatrix}, \quad P_3 = \frac{1}{3\Delta^2} \left\{ \begin{vmatrix} R_1^3 & R_2^3 & R_3^3 \\ \Delta_1^3 & \Delta_2^3 & \Delta_3^3 \\ 1 & 1 & 1 \end{vmatrix} + \prod_{1 \leq m < n \leq 3} (R_n\Delta_n - R_m\Delta_m) \right\}. \tag{30}$$

Formula (28) can be proved at least in two ways. The first is based on the 1D Fourier image of the product of two Airy functions ([36] Section 3.5.3): If

$$F(x) = \int_{\mathbb{R}} e^{-ixt} \text{Ai}(A_1 t + B_1)\text{Ai}(A_2 t + B_2) \, dt, \tag{31}$$

then

$$F(x) = \frac{1}{|A_1^3 - A_2^3|^{1/3}} \exp\left(\frac{i(A_1^2 B_1 - A_2^2 B_2)x}{A_1^3 - A_2^3} + \frac{i(A_1^3 + A_2^3)x^3}{3(A_1^3 - A_2^3)^2}\right) \times \tag{32}$$

$$\times \text{Ai}\left(\frac{1}{(A_1^3 - A_2^3)^{1/3}}\left[A_1 B_2 - A_2 B_1 - \frac{A_1 A_2 x^2}{A_1^3 - A_2^3}\right]\right), \qquad A_1 \neq A_2$$

$$F(x \neq 0) = \frac{1}{2\sqrt{\pi|Ax|}} \exp\left(\frac{ix^3}{12A^3} + \frac{i(B_2 + B_1)x}{2A} - \frac{iA(B_2 - B_1)^2}{4x} + \frac{\pi i}{4}\,\text{sgn}(Ax)\right), \quad A_1 = A_2 = A \neq 0$$

$$F(0) = \frac{1}{|A|}\,\delta(B_2 - B_1). \qquad A_1 = A_2 = A \neq 0$$

The second way is similar to the proof of Equation (3) in Appendix A but lengthy. Nevertheless, both ways are straightforward. Moreover, a desire to find the representations of the polynomials $P_n$ in (28) which demonstrate their invariance under permutations of the triples $\{A_n, B_n, C_n\}$, requires quite tedious algebra.

The formula (28) has numerous and important corollaries. Some of them, interesting for mathematics, are mentioned in [37]. For optics, the light field $\text{Ai}_3(\mathbf{r} \,|\, \mathbf{M})$ is a field of finite or infinite energy provided $\Delta > 0$ or $\Delta < 0$, respectively. The condition $\Delta < 0$ can be expressed in terms of the relative position of the vectors $(A_n, B_n)$, $1 \leq n \leq 3$, on the plane: If there is a straight line passing through the origin such that all three vectors are located along one side from it, then $\Delta < 0$. And vice versa, if such a line does not exist, then $\Delta > 0$.

Consider an example: Let the product of Airy functions in Equation (26) be as follows:

$$\text{Ai}_3(\mathbf{r}\,|\,\mathbf{M}) = \text{Ai}(-x\sin\alpha + y\cos\alpha + C_1)\text{Ai}(x\sin\alpha + y\cos\alpha + C_2)\text{Ai}(y + C_3). \tag{33}$$

Then

$$(A_1, B_1) = (-\sin\alpha, \cos\alpha), \quad (A_2, B_2) = (\sin\alpha, \cos\alpha), \quad (A_3, B_3) = (0, 1),$$
$$\Delta_1 = -2\cos\alpha\sin\alpha, \quad \Delta_2 = \Delta_3 = \sin\alpha, \quad \Delta = 2\sin^3\alpha(1 - 4\cos^3\alpha),$$
$$R_1 = x\cos\alpha + y\sin\alpha, \quad R_2 = x\cos\alpha - y\sin\alpha, \quad R_3 = x, \tag{34}$$

and the polynomials $P_n$ are reduced to simpler expressions:

$$P_0 = \frac{C_1 + C_2 - 2C_3\cos\alpha}{2^{1/3}(1 - 4\cos^3\alpha)^{1/3}},$$
$$P_1 = \frac{C_2 - C_1}{2\sin\alpha}\cdot x + \frac{C_3 - 2(C_1 + C_2)\cos^2\alpha}{1 - 4\cos^3\alpha}\cdot y,$$
$$P_2 = \frac{1}{2^{1/3}(1 - 4\cos^3\alpha)^{1/3}}\left(\frac{1}{2\sin^2\alpha}\cdot x^2 - \frac{2\cos\alpha}{1 - 4\cos^3\alpha}\cdot y^2\right),$$
$$P_3 = -\frac{\cos^2\alpha}{\sin^2\alpha(1 - 4\cos^3\alpha)}\cdot x^2 y + \frac{1 + 4\cos^3\alpha}{3(1 - 4\cos^3\alpha)^2}\cdot y^3. \tag{35}$$

One of the most interesting cases is when the RHS of Equation (28) contains a circular Airy factor. For this, the polynomial $P_2$ should be radially symmetric, which leads to the equation

$$\frac{1}{2\sin^2\alpha} = -\frac{2\cos\alpha}{1 - 4\cos^3\alpha} \quad \Rightarrow \quad 8\cos^3\alpha - 4\cos\alpha - 1 = 0. \tag{36}$$

Introducing the variable $t = 2\cos\alpha$, one obtains the cubic equation $t^3 - 2t - 1 = 0$. Its roots are $t = \{-1, (1\pm\sqrt{5})/2\}$ that corresponds to $\alpha = \{2\pi/3, \pi/5, 3\pi/5\}$. We discuss these three cases separately.

(i) If $\alpha = 2\pi/3$, then $\Delta = 3^{5/2}/2^3$, the field (33) is of finite energy, and

$$\mathcal{F}\big[\text{Ai}_3(\boldsymbol{\rho}\,|\,\mathbf{M})\big](\mathbf{r}) = \frac{1}{3^{5/6}\pi}\exp\left(\frac{i}{3}\{\sqrt{3}(C_2 - C_1)x - (C_1 + C_2 - 2C_3)y\}\right.$$
$$\left. - \frac{2i}{27}(3x^2 y - y^3)\right)\text{Ai}\left(\frac{C_1 + C_2 + C_3}{3^{1/3}} + \frac{2}{3^{4/3}}(x^2 + y^2)\right). \tag{37}$$

When $C_1 = C_2 = C_3 = c$, this formula coincides with Equation (3).

(ii) If $\alpha = \pi/5$, then

$$\cos\alpha = \frac{1 + \sqrt{5}}{4}, \quad \sin\alpha = \frac{1 + \sqrt{5}}{4}\sqrt{5 - 2\sqrt{5}}, \quad \Delta = -\sqrt{5}\sin^3\alpha < 0, \tag{38}$$

and the field (33) is of infinite energy. Its Fourier image is

$$\mathcal{F}\big[\text{Ai}_3(\boldsymbol{\rho}\,|\,\mathbf{M})\big](\mathbf{r}) = \frac{\sqrt{10 + 2\sqrt{5}}}{5^{2/3}2\pi}\exp(iP_1 + iP_3)\text{Ai}(P_0 + P_2), \tag{39}$$
$$P_0 = -\frac{1}{5^{1/6}}\left(C_1 + C_2 - \frac{1 + \sqrt{5}}{2}C_3\right),$$
$$P_1 = \sqrt{\frac{5 + \sqrt{5}}{10}}(C_2 - C_1)\cdot x + \frac{(3 + \sqrt{5})(C_1 + C_2) - 4C_3}{2\sqrt{5}}\cdot y,$$
$$P_2 = -\frac{1 + \sqrt{5}}{5^{2/3}}(x^2 + y^2), \quad P_3 = \frac{2(2 + \sqrt{5})}{5}\cdot x^2 y + \frac{2(4 + \sqrt{5})}{15}\cdot y^3.$$

Since $P_2$ is negative, the field (39) has an infinite number of zero circles. Figure 9 depicts an example when $C_1 = C_2 = c$, $C_3 = \frac{1}{2}(3 + \sqrt{5})c$, and $5^{1/3}c = a'_1$. Then, the polynomial $P_1$ vanishes, and the intensity becomes maximum at the origin.

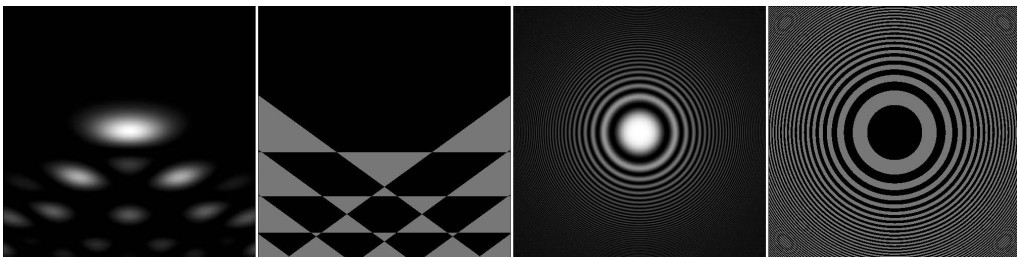

**Figure 9.** Numerically evaluated intensity and phase (without cubic component) distributions of the three-Airy beam of infinite energy and of its Fourier image (from left to right). All frames are shown in the square $[-h, h] \times [-h, h]$, where $h = 4.5$.

(iii)   If $\alpha = 3\pi/5$, then

$$\cos\alpha = -\frac{\sqrt{5} - 1}{4}, \quad \sin\alpha = \frac{\sqrt{5} - 1}{4}\sqrt{5 + 2\sqrt{5}}, \quad \Delta = \sqrt{5}\sin^3\alpha > 0, \quad (40)$$

and the field (33) is of finite energy. Its Fourier image is

$$\mathcal{F}\left[\mathrm{Ai}_3(\boldsymbol{\rho} \,|\, \mathbf{M})\right](\mathbf{r}) = \frac{\sqrt{10 - 2\sqrt{5}}}{5^{2/3}2\pi}\exp(\mathrm{i}P_1 + \mathrm{i}P_3)\mathrm{Ai}(P_0 + P_2), \quad (41)$$

$$P_0 = \frac{1}{5^{1/6}}\left(C_1 + C_2 + \frac{\sqrt{5} - 1}{2}C_3\right),$$

$$P_1 = \sqrt{\frac{5 - \sqrt{5}}{10}}(C_2 - C_1)\cdot x - \frac{(3 - \sqrt{5})(C_1 + C_2) - 4C_3}{2\sqrt{5}}\cdot y,$$

$$P_2 = \frac{\sqrt{5} - 1}{5^{2/3}}(x^2 + y^2), \quad P_3 = -\frac{2(\sqrt{5} - 2)}{5}\cdot x^2 y + \frac{2(4 - \sqrt{5})}{15}\cdot y^3.$$

The polynomial $P_2$ is positive, so the field (41) has a finite number of zero circles. As before, we choose the coefficients $C_n$ in such a way as to vanish the polynomial $P_1$: $C_1 = C_2 = c$, $C_3 = \frac{1}{2}(3 - \sqrt{5})c$. When $5^{1/3}c = a'_1, a_1, a'_4, a_4$, the initial three-Airy beams, and their Fourier images are depicted in Figure 10.

Finally, let us consider the field (33) for the limiting case $\alpha \to +0$. Then $\sin\alpha \cong \alpha$, $\cos\alpha \cong 1$, $\Delta \cong -6\alpha^3$, and

$$P_0 = \frac{2C_3 - C_1 - C_2}{6^{1/3}}, \quad P_1 = \frac{C_2 - C_1}{2\alpha}\cdot x + \frac{2(C_1 + C_2) - C_3}{3}\cdot y,$$

$$P_2 = -\frac{1}{6^{4/3}}\left(\frac{3x^2}{\alpha^2} + 4y^2\right), \quad P_3 = \frac{x^2 y}{3\alpha^2} + \frac{5y^3}{27}. \quad (42)$$

Substituting these expressions into Equation (28) and integrating both sides over $x \in \mathbb{R}$ (at his stage, the integral over $\xi$ reduces to Dirac's delta function and $\alpha$ vanishes), we have

$$2\pi\int_{\mathbb{R}}\mathrm{e}^{-\mathrm{i}y\eta}\prod_{1 \leq n \leq 3}\mathrm{Ai}(\eta + C_n)\,\mathrm{d}\eta = \frac{1}{6^{1/3}}\exp\left[\frac{\mathrm{i}(2C_1 + 2C_2 - C_3)y}{3} + \frac{5\mathrm{i}y^3}{27}\right]$$

$$\times \int_{\mathbb{R}}\exp\left[\frac{\mathrm{i}(C_2 - C_1)x}{2} + \frac{\mathrm{i}x^2 y}{3}\right]\mathrm{Ai}\left(\frac{2C_3 - C_1 - C_2}{6^{1/3}} - \frac{3x^2 + 4y^2}{6^{4/3}}\right)\mathrm{d}x. \quad (43)$$

This quite unusual identity is applicable in the theory of hypergeometric functions and helps to investigate the elliptic umbilic diffraction catastrophe [26], and has various

corollaries. We provide only one of them, the integral representation of the product of both Airy functions (see also Section 3.6.3 in [36]):

$$\text{Ai}(c)\text{Bi}(c) = 3^{1/6} \int_{\mathbb{R}} \text{Ai}^2(\eta)\text{Ai}(\eta + 3^{1/3}c)\,d\eta. \tag{44}$$

To prove this, we substitute $y = 0$ into (43), then the integral over $x$ is known due to Berry ([31], Equation (B17)). The formula (44) helps to study the propagation of certain Gaussian and non-Gaussian light fields of finite and infinite energy containing the factor AiBi.

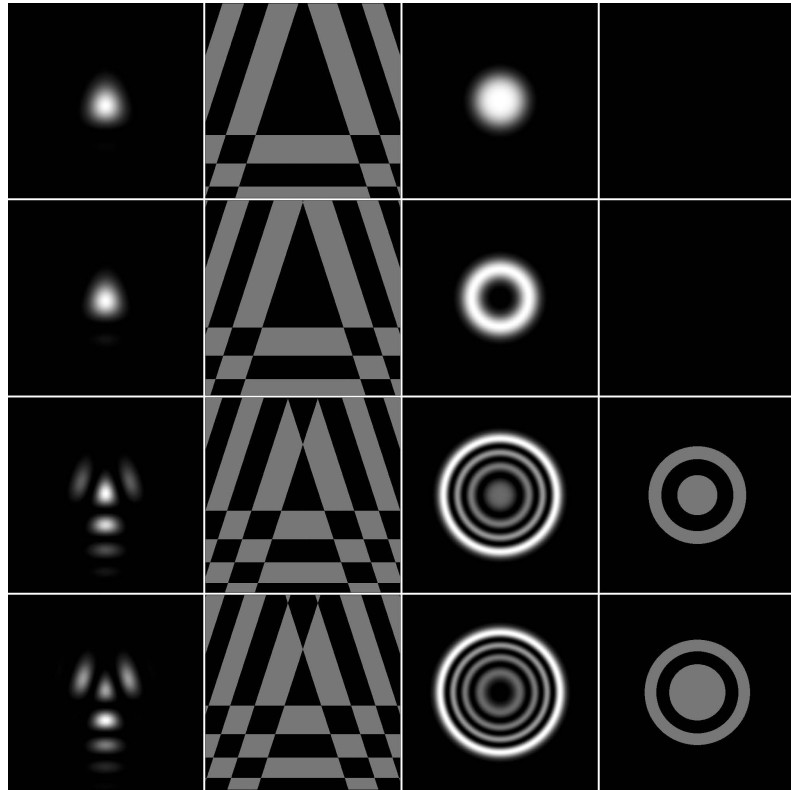

**Figure 10.** Numerically evaluated intensity and phase (without cubic component) distributions of the three-Airy beam of finite energy and of its Fourier image (from left to right) for the shift parameters $5^{1/3}c = a'_1, a_1, a'_4, a_4$ (from top to bottom). All frames are shown in the square $[-h, h] \times [-h, h]$, where $h = 4.5$.

## 5. Conclusions

The problems studied above related to the propagation of three-Airy beams in the Fresnel zone have close connections with hot topics in optics. We believe that generalized three-Airy beams will be useful for various applications of modern optics and photonics, such as laser structuring of photosensitive material surfaces and laser ablation [38,39], optical trapping and manipulation [40,41].

For the theory, the results obtained can be used as a basis for studying other light fields. For example, when talking about Airy laser beams, we usually mean the beams constructed for the function $\text{Ai}(x)$, ignoring the function $\text{Bi}(x)$. The only exception known to the authors is Ref. [42], in which the study of the initial light field $F_0(x) = \exp(-x^2/w^2)\text{Bi}(Ax + B)$ was conducted by Torre. It is quite easy to transfer all the results obtained for Airy–Gaussian beams in [3] (here, Airy means Ai) to the case of Bi. One of the possible ways is to apply the fact that both functions, $\text{Ai}(x)$ and $\text{Bi}(x)$, are solutions of the differential equation, $y'' = xy$. The second way is to use algebraic relations ([43], Equations (10.4.6) and (10.4.9)),

$$2\omega\text{Ai}(\omega x) = -\text{Ai}(x) + \mathrm{i} \cdot \text{Bi}(x), \quad 2\omega\text{Bi}(\omega x) = -\text{Bi}(x) + 3\mathrm{i} \cdot \text{Ai}(x), \tag{45}$$

where $\omega = e^{2\pi i/3}$ is a cubic root of unity.

For the initial field $F_0(x) = \exp(-x^2/w^2)\{\text{Ai} + \text{i} \cdot \text{Bi}\}(Ax + B)$, it is also easy to construct a solution of the paraxial equation. It can be seen that the resulting beam, $F(x, z)$, exhibits the property of "bending the trajectory" as it propagates.

In addition, the connection between Airy functions with Bessel functions of orders $\pm 1/3$ makes it possible to construct fractional Bessel–Gaussian (BG) beams of these orders. These beams are quite different from the BG beams of integer and half-integer orders known before [44–46]. It would be especially interesting to study three-Airy beams of infinite energy, i.e., fractional-order Bessel beams, to reveal the effect of larger depth of focus [47]. Moreover, replacing Airy functions in the initial fields above with the Pearcey and Swallowtail functions leads to light fields related to Bessel functions of the order of $\pm 1/4$ and $\pm 1/5$.

Finally, the study of the propagation of generalized three-Airy beams $\text{Ai}_3(\mathbf{r} \mid \mathbf{M})$ in the Fresnel zone requires the use of diffraction catastrophe integrals of order higher than 3. For example, considering Equation (28) for the general case of $P_2$ instead of the radially symmetric case $P_2(\mathbf{r}) = cr^2$, one can obtain

$$\int_{\mathbb{R}} e^{ix\xi} \text{Ai}(2^{2/3}\xi) \text{Ai}(\xi + C_1) \text{Ai}(\xi + C_2) \, d\xi = \frac{|x|^{-1/4}}{2^{5/3}\pi^{3/2}} \exp\left(\frac{ixX^2}{4} - \frac{ix^3}{12} - \frac{\pi i}{4} \operatorname{sgn} x\right)$$

$$\times \begin{cases} \text{Pe}\left[(C_2 - C_1)|x|^{1/4}, -X|x|^{1/4}\right], & (x > 0), \\ \text{Pe}^*\left[-(C_2 - C_1)|x|^{1/4}, X|x|^{1/4}\right], & (x < 0), \end{cases} \quad (46)$$

where $X = x/2 - (C_1 + C_2)/x$, the asterisk means complex conjugation, and

$$\text{Pe}(x, y) = \int_{\mathbb{R}} \exp\left(it^4 + iyt^2 + ixt\right) dt \quad (47)$$

is the Pearcey function [48,49], i.e., the diffraction catastrophe of order 4. In particular,

$$\int_{\mathbb{R}} e^{ix\xi} \text{Ai}(2^{2/3}\xi) \text{Ai}^2\left(\xi + \frac{x^2}{4}\right) d\xi = \frac{\Gamma(5/4)|x|^{-1/4}}{2^{2/3}\pi^{3/2}} \exp\left(-\frac{ix^3}{12} - \frac{\pi i}{8} \operatorname{sgn} x\right), \quad (48)$$

since $\text{Pe}(0, 0) = 2\Gamma(5/4)e^{\pi i/8}$. This result confirms a very important statement given in [50]: Fourier and Fresnel transforms of light fields constructed using diffraction catastrophes of order $n$ sometimes require the use of diffraction catastrophes whose order exceeds $n$.

**Author Contributions:** Conceptualization, E.G.A.; methodology, E.G.A. and S.N.K.; software, E.G.A. and S.N.K.; validation, E.G.A., S.N.K. and R.V.S.; formal analysis, S.N.K.; investigation, E.G.A., S.N.K. and R.V.S.; resources, E.G.A. and S.N.K.; data curation, S.N.K. and R.V.S.; writing—original draft preparation, E.G.A.; writing—review and editing, E.G.A., S.N.K. and R.V.S.; visualization, E.G.A., S.N.K. and R.V.S. All authors have read and agreed to the published version of the manuscript.

**Funding:** This work was supported by the Russian Science Foundation under grant 23-22-00314 (theory and numerical simulation) and within the state assignment of NRC "Kurchatov Institute" (experiment).

**Institutional Review Board Statement:** Not applicable.

**Informed Consent Statement:** Not applicable.

**Data Availability Statement:** Data are contained within the article.

**Acknowledgments:** We thank Tatiana Alieva, who pointed out that the functional Fresnel transform is reduced to a fractional Fourier transform and brought to our attention an important paper on the subject.

**Conflicts of Interest:** The authors declare no conflict of interest.

## Appendix A. Evaluation of the Fourier Image of Three-Airy Beams

Here, we present a proof of Equation (3), which is considerably simpler than that proposed in [19]. Below, we use dimensionless variables and set $b = 1$ for brevity.

Let us write the beam $\text{Ai}_3$ as the product of three copies of the Airy function on the base of the integral representation (2) and regroup the linear terms depending on $x$, $y$, $c$:

$$\text{Ai}_3(\mathbf{r}\,|\,c) = \frac{1}{(2\pi)^3} \iiint_{\mathbb{R}^3} \exp\left[\frac{i}{3}(\xi^3 + \eta^3 + \zeta^3)\right.$$
$$\left. + ix \cdot \frac{\sqrt{3}}{2}(\xi - \eta) + iy\left(\zeta - \frac{\xi + \eta}{2}\right) + ic(\xi + \eta + \zeta)\right] d\xi\, d\eta\, d\zeta. \quad \text{(A1)}$$

Replacing variables $(\xi, \eta, \zeta) \to (u, v, w)$ by the relations

$$\begin{pmatrix} u \\ v \\ w \end{pmatrix} = \frac{1}{2}\begin{pmatrix} \sqrt{3} & -\sqrt{3} & 0 \\ -1 & -1 & 2 \\ 2 & 2 & 2 \end{pmatrix}\begin{pmatrix} \xi \\ \eta \\ \zeta \end{pmatrix} \quad \Rightarrow \quad \begin{pmatrix} \xi \\ \eta \\ \zeta \end{pmatrix} = \frac{1}{3}\begin{pmatrix} \sqrt{3} & -1 & 1 \\ -\sqrt{3} & -1 & 1 \\ 0 & 2 & 1 \end{pmatrix}\begin{pmatrix} u \\ v \\ w \end{pmatrix}, \quad \text{(A2)}$$

we obtain

$$\text{Ai}_3(\mathbf{r}\,|\,c) = \frac{1}{(2\pi)^3} \cdot \frac{2}{3\sqrt{3}} \iiint_{\mathbb{R}^3} \exp\left[i(xu + yv + cw) + \frac{i}{27}P(u,v,w)\right] du\, dv\, dw, \quad \text{(A3)}$$

where $P(u,v,w) = 2v^3 - 6u^2v + 6u^2w + 6v^2w + w^3$ is a homogeneous cubic polynomial. The integral of $dw$ is reduced to the Airy function by the change $w = 3^{2/3}t$, and we obtain

$$\text{Ai}_3(\mathbf{r}\,|\,c)$$
$$= \frac{1}{2\pi^2\, 3^{5/6}} \iint_{\mathbb{R}^2} \exp\left[i(xu + yv) + \frac{2i}{27}(v^3 - 3u^2v)\right] \text{Ai}\left[3^{2/3}c + \frac{2}{3^{4/3}}(u^2 + v^2)\right] du\, dv.$$
$$\text{(A4)}$$

Recognizing this integral as the inverse 2D Fourier transform $(u, v) \to (x, y)$, we obtain the formula, which is the same as the relation (3).

It should be noted that the attempt to fit the integral of $dv$ into an Airy function shape leads to the Fourier transform in terms $(u, w) \to (x, c)$, which reduces to the trivial identity, $\text{Ai}_3 = \text{Ai}_3$, due to Berry's integrals (see Equations (B17) and (B18) in [31]).

## Appendix B. Fractional Fourier Transform for Visualization Purposes of a Light Beam Propagation in the Fresnel Zone

It is well known that Hermite–Gaussian (HG) modes,

$$\mathcal{H}_{n,m}(\mathbf{r}) = \exp(-|\mathbf{r}|^2)H_n(\sqrt{2}\,x)H_m(\sqrt{2}\,y), \quad (n, m, = 0, 1, \ldots), \quad \text{(A5)}$$

keep their structural stability upon propagation in the Fresnel diffraction zone:

$$\mathbf{FR}_z\left[\mathcal{H}_{n,m}\left(\frac{\boldsymbol{\rho}}{w}\right)\right](\mathbf{r}) = \frac{1}{\sigma}\exp\left(\frac{2iz|\mathbf{r}|^2}{kw^4|\sigma|^2} - i(n + m)\arg\sigma\right)\mathcal{H}_{n,m}\left(\frac{\mathbf{r}}{w|\sigma|}\right). \quad \text{(A6)}$$

Here $w$ is a Gaussian spot width, and $\sigma = 1 + 2iz/kw^2$ is an auxiliary complex parameter. Thus, as $z$ grows from 0 to $+\infty$, the HG mode changes in scale, $\mathbf{r} \to \mathbf{r}/|\sigma|$, and two pure phase factors appear. The first one is a defocusing factor, $\exp(2iz|\mathbf{r}|^2/kw^4|\sigma|^2)$. The second, $\exp(-i(n + m)\arg\sigma)$, demonstrates the phase shift of the beam during propagation and depends on the Gouy phase.

If $F_0(\mathbf{r})$ is a coherent light field of finite energy, then it is well known that it can be expanded into a series of HG modes,

$$F_0(\mathbf{r}) = \sum_{n,m \geq 0} c_{n,m} \mathcal{H}_{n,m}\left(\frac{\mathbf{r}}{w}\right), \tag{A7}$$

where the coefficients are easy to find,

$$c_{n,m} = \frac{1}{\pi\, 2^{n+m-1} n!\, m!} \iint_{\mathbb{R}^2} F_0(\boldsymbol{\rho}) \mathcal{H}_{n,m}\left(\frac{\boldsymbol{\rho}}{w}\right) \frac{\mathrm{d}^2\boldsymbol{\rho}}{w^2}, \tag{A8}$$

due to the orthogonality property of HG modes:

$$\iint_{\mathbb{R}^2} \mathcal{H}_{n,m}(\mathbf{r}) \mathcal{H}_{\nu,\mu}(\mathbf{r})\, \mathrm{d}^2\mathbf{r} = \pi\, 2^{n+m-1} n!\, m!\, \delta_{n,\nu} \delta_{m,\mu}. \tag{A9}$$

Thus, for a light field, defined by its complex amplitude distribution $F_0(\mathbf{r})$ in the initial plane $z = 0$, the field evolution in the Fresnel zone is reduced to calculating a series similar to the series (A7):

$$\mathbf{FR}_z\big[F_0(\boldsymbol{\rho})\big](\mathbf{r}) = \frac{1}{\sigma} \exp\left(\frac{2\mathrm{i}z|\mathbf{r}|^2}{kw^4|\sigma|^2}\right) \sum_{n,m \geq 0} c_{n,m} \mathrm{e}^{-\mathrm{i}(n+m)\arg\sigma} \mathcal{H}_{n,m}\left(\frac{\mathbf{r}}{w|\sigma|}\right). \tag{A10}$$

For any number of values of $z$, this approach requires a single calculation of the coefficients (A8). Thus, this method of calculating the Fresnel transform is very time-efficient for some families of light fields (see [19,51]). Nevertheless, finding the optimal parameter $w$ and the ranges $0 \leq n \leq N$, $0 \leq m \leq M$ for the double series truncation may turn out to be a nontrivial problem, which strongly depends on the properties of the initial field $F_0(\mathbf{r})$.

Let us now assume that we want to show the change in the field $F_0(\mathbf{r})$ as it propagates in the Fresnel zone at $z \in [0, \infty]$. For this, we need to calculate a sequence of transverse distributions of intensity and phase at discrete points $\{z_0 = 0, z_1, z_2, \ldots, z_N = \infty\}$. It seems reasonable when the sequence of such frames is generated by a computer to use a *functional Fresnel transform*, removing the defocusing factor and the scaling that occurs during the propagation of the beam (A10):

$$\mathbf{funcFR}_{z,w}\big[F_0(\boldsymbol{\rho})\big](\mathbf{r}) = \sum_{n,m \geq 0} c_{n,m} \mathrm{e}^{-\mathrm{i}(n+m)\arg\sigma} \mathcal{H}_{n,m}\left(\frac{\mathbf{r}}{w}\right). \tag{A11}$$

Here, unlike the Fresnel transform, we have added the parameter $w$ as an index to emphasize that the functional Fresnel transform depends not only on the choice of the plane $z$, but also on the choice of the Gaussian width $w$:

$$\mathbf{funcFR}_{z,w}\big[F_0(\boldsymbol{\rho})\big](\mathbf{r}) \overset{(A10)}{=} \sigma \exp\left(-\frac{2\mathrm{i}z|\mathbf{r}|^2}{kw^4}\right) \mathbf{FR}_z\big[F_0(\boldsymbol{\rho})\big](|\sigma|\mathbf{r}). \tag{A12}$$

Replacing (A10) with (A11) solves two visualization problems. The first is to reduce moire on the phase distribution (the moire effect is typical for the phase of the field (A10) due to oscillations that become increasingly stronger as $z$ grows. The second is to eliminate the intensity magnifying scaling of the field (A10) in the $x, y$ plane as $z$ grows, since otherwise, with fixed frame sizes, significant details of the field would begin to go beyond the frame boundaries.

Applying (A11) to a single HG mode with a Gaussian width $w_1$, one obtains

$$\textbf{funcFR}_{z,w}\left[\mathcal{H}_{n,m}\left(\frac{\boldsymbol{\rho}}{w_1}\right)\right](\mathbf{r})$$

$$= \frac{\sigma}{\sigma_1}\exp\left(\frac{2\mathrm{i}z|\mathbf{r}|^2}{k|\sigma_1|^2}\left[\frac{1}{w_1^4}-\frac{1}{w^4}\right]-\mathrm{i}(n+m)\arg\sigma_1\right)\mathcal{H}_{n,m}\left(\frac{\mathbf{r}}{w_1}\cdot\frac{|\sigma|}{|\sigma_1|}\right), \qquad (A13)$$

where $\sigma_1 = 1 + 2\mathrm{i}z/kw_1^2$. If $w_1 = w$, then the intensity of the field in the RHS of Equation (A13) is *the same* for all $z$:

$$\textbf{funcFR}_{z,w}\left[\mathcal{H}_{n,m}\left(\frac{\boldsymbol{\rho}}{w}\right)\right](\mathbf{r}) = \mathrm{e}^{-\mathrm{i}(n+m)\arg\sigma}\mathcal{H}_{n,m}\left(\frac{\mathbf{r}}{w}\right). \qquad (A14)$$

If $w_1 \neq w$, then the phase defocusing factor and intensity scaling are present in (A13) but remain limited for all $z \in [0, +\infty]$. Moreover, the intensity of the field in the RHS of (A13) remains the same at $z = \infty$ as at $z = 0$, regardless of the values of $w_1$ and $w$.

The relationship (A14) demonstrates that HG modes are eigenfunctions for the operator $\textbf{funcFR}_{z,w}$, corresponding to the eigenvalues $\mathrm{e}^{-\mathrm{i}(n+m)\arg\sigma}$. This equality is very similar to the well-known property of HG modes,

$$\mathcal{F}_\gamma\left[\mathcal{H}_{n,m}\left(\frac{\boldsymbol{\rho}}{\sqrt{2}}\right)\right](\mathbf{r}) = \mathrm{e}^{-\mathrm{i}(n+m)\gamma}\,\mathcal{H}_{n,m}\left(\frac{\mathbf{r}}{\sqrt{2}}\right), \qquad (A15)$$

where

$$\mathcal{F}_\gamma[f(\boldsymbol{\rho})](\mathbf{r}) = \frac{\mathrm{e}^{\mathrm{i}\gamma}}{2\pi\mathrm{i}\sin\gamma}\iint_{\mathbb{R}^2}\exp\left(-\mathrm{i}\frac{\langle\mathbf{r},\boldsymbol{\rho}\rangle}{\sin\gamma}+\mathrm{i}\frac{|\mathbf{r}|^2+|\boldsymbol{\rho}|^2}{2\tan\gamma}\right)f(\boldsymbol{\rho})\,\mathrm{d}^2\boldsymbol{\rho} \qquad (A16)$$

is the fractional Fourier transform, $\gamma \in [0, \pi/2]$. This suggests that the functional Fresnel transform and the fractional Fourier transform are closely related. Indeed, the relationship between these operators can be obtained on the basis of Equation (A12).

It is well known that ordinary Fresnel transform and fractional Fourier transform have much in common [52,53]. More precisely, there is a one-to-one correspondence between $\textbf{FR}_z[F_0(\boldsymbol{\rho})](\mathbf{r})$ for $z \in [0, \infty)$ and $\mathcal{F}_\gamma[F_0(\boldsymbol{\rho})](\mathbf{r})$ for $\gamma \in [0, \pi/2]$: If $F(\mathbf{r}, z) = \textbf{FR}_z[F_0(\boldsymbol{\rho})](\mathbf{r})$ is known, then (below we use dimensionless vectors $\mathbf{r}$, $\boldsymbol{\rho}$ in both transforms)

$$\mathcal{F}_\gamma[F_0(\boldsymbol{\rho})](\mathbf{r}) = \frac{\mathrm{e}^{\mathrm{i}\gamma}}{\cos\gamma}\exp\left(-\frac{\mathrm{i}|\mathbf{r}|^2\tan\gamma}{2}\right)F\left(\frac{w\mathbf{r}}{\sqrt{2}\cos\gamma},\frac{kw^2}{2}\tan\gamma\right). \qquad (A17)$$

And vice versa: If $\mathcal{F}_\gamma[F_0(\boldsymbol{\rho})](\mathbf{r})$ is known, then

$$F\left(\frac{w\mathbf{r}}{\sqrt{2}},z\right) = \frac{1}{\sigma}\exp\left[\frac{(\sigma-1)|\mathbf{r}|^2}{2|\sigma|^2}\right]\mathcal{F}_{\arg\sigma}\left[F_0\left(\frac{w\boldsymbol{\rho}}{\sqrt{2}}\right)\right]\left(\frac{\mathbf{r}}{|\sigma|}\right), \qquad (A18)$$

where $\sigma = 1 + 2\mathrm{i}z/kw^2$. As result,

$$\textbf{funcFR}_{z,w}[F_0(\boldsymbol{\rho})](\mathbf{r}) \stackrel{(A12),(A18)}{=} \mathcal{F}_\gamma\left[F_0\left(\frac{w\boldsymbol{\rho}}{\sqrt{2}}\right)\right]\left(\frac{\mathbf{r}\sqrt{2}}{w}\right). \qquad (A19)$$

Thus, if we want to demonstrate the propagation of the field $F_0(\mathbf{r})$ based on the expression (A11) instead of (A10), we need to use the fractional Fourier transform instead of the Fresnel transform. In Figure 1, we use equality (A19) to visualize the transverse intensity and phase distributions at equidistant sample points: $\gamma_n = \arctan z_n = n\pi/2N$, where $n = 0, 1, \ldots, N$.

And finally, we mention that Equation (A17) allows the discovery of the fractional Fourier transform of any initial field $F_0(\mathbf{r})$ if the corresponding solution $F(\mathbf{r}, z)$ of the paraxial equation is known. In particular,

$$\mathcal{F}_\gamma\left[\mathcal{H}_{n,m}\left(\frac{\boldsymbol{\rho}}{w}\right)\right](\mathbf{r}) = \frac{\mathrm{e}^{\mathrm{i}\gamma}}{\tilde{\sigma}} \exp\left(\frac{\mathrm{i}|\mathbf{r}|^2 \sin 2\gamma}{4|\tilde{\sigma}|^2}\left[\frac{4}{w^4} - 1\right] - \mathrm{i}(n+m)\arg\tilde{\sigma}\right)\mathcal{H}_{n,m}\left(\frac{\mathbf{r}}{w|\tilde{\sigma}|}\right), \quad \text{(A20)}$$

where $\tilde{\sigma} = \cos\gamma + \mathrm{i}(2/w^2)\sin\gamma$ is an auxiliary parameter, similar to the parameter $\sigma$, which appeared in (A6).

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
