# Peer review of "Three-Airy Beams, Their Propagation in the Fresnel Zone, the Autofocusing Plane Location, as Well as Generalizing Beams"

_photonics, doi:10.3390/photonics11040312_

Round 1

Reviewer 1 Report

Comments and Suggestions for Authors

The authors reported a study of the propagation of three-Airy beams in the Fresnel zone by theoretical, numerical and experimental means. The ways to evaluate the Fresnel transform of these beams were discussed. The experimental implementation of three-Airy beams was carried out. An autofocusing behaviour of these beams was investigated. Some corollaries concerning the three-Airy beams of finite and infinite energy were provided. The obtained results showed that the autofocusing plane of a three-Airy beam is similar to square root of the shift parameter. This work may find potential applications in optical trapping and manipulation. As far as I can see all of the theory, numerical analysis, and experiment are sound. I recommend this paper to be published in Photonics. Here are two suggestions:

1. The introduction section can be written more concisely. There is no need to include so many formulas in the introduction.

2. The figures are not standardized enough. At least one of the sub-figures in each figure needs to be marked with a coordinate axis and unit.

Author Response

1. The introduction section can be written more concisely. There is no need to include so many
formulas in the introduction.
We have slightly changed the Introduction by removing the integral definition of the function
Ai(x) from the first page. However, the three-Airy beam is a quite difficult object to study,
and success here is hardly possible without the help of mathematics. Of course, we could limit
ourselves in the Introduction to a purely verbal discussion, but it seems to us that such an
approach would deceive the reader with the false simplicity of the object under investigation,
rather than prepare him for the need to use integral transforms and complex analysis. In this
sense, the papers of Michael Berry is the best example for us.
We tried to present the results as simply as possible, sometimes repeating well-known facts
instead of providing links. In addition, we preferred to demonstrate the main object of our
research right away, so that a thoughtful reader working in optics can immediately conclude
whether this object is interesting to him or not.
2. The figures are not standardized enough. At least one of the sub-figures in each figure needs to
be marked with a coordinate axis and unit.
We have added the following text to the caption of the Figure 1:
All frames are shown in the square [−h, h] × [−h, h], where h = 4.0 for the top row and h = 4.5
for the bottom. As usual, the x axis is horizontal, the y axis is vertical.
Besides, we have changed the captions of Figures 9 and 10 in the same manner.

Reviewer 2 Report

Comments and Suggestions for Authors

It was my pleasure to read the submission. My opinion of this submission is very positive and recommend a minor revision, with following comments.

1.Speaking about the far zone, the authors call it the Fourier zone, which does not seem very good to me, although passable.

2. In my opinion, formula (7) requires a reference and some comment. I have not noticed any mention of the parabolic-equation approach.

3. Theoretical and experimental results should be more clearly distinguished in the figure captions.

4. It seems that the authors consider the wavefield not exclusively in the Fresnel zone, as indicated in the title.

Author Response

1. Speaking about the far zone, the authors call it the Fourier zone, which does not seem very good to me, although passable.
We have replaced ‘the Fourier zone’ with ’the far zone’ in the Abstract.
2. In my opinion, formula (7) requires a reference and some comment. I have not noticed any mention of the parabolic-equation approach.
We have added the 2D paraxial equation and the reference just before Eq.(7).
. . . in the paraxial approximation by the equation ${\partial_x^2+\partial_y^2+2ik\partial_z\}F(r,z)= 0, where
F(r, 0) = F_0(r). If F_0(r) vanishes sufficiently rapidly as $|r| \to\infty$, then F(r, z) related with F_0(r) by the Fresnel transform [Goodman]:
3. Theoretical and experimental results should be more clearly distinguished in the figure captions.
We have changed the captions of Figures 1, 3, 9, and 10. All of them are result of numerical evaluation.
4. It seems that the authors consider the wavefield not exclusively in the Fresnel zone, as indicated in the title.
Yes, some parts of our paper are devoted to other issues. With some regret, we have changed the title of our article. Now it has become longer, although it more accurately reflects the issues discussed.
Three-Airy beams, their propagation in the Fresnel zone, the autofocusing plane location, as well as generalizing beams

Reviewer 3 Report

Comments and Suggestions for Authors

This is an interesting paper to read, though I have some comments regarding a) introduction, b) flow of the paper and c) formulas.

Introduction could be smoother, not all readers are skilled in the mathematics, so the first formula comming in the Introduction at the third line is a hard to swallow. 

Next, I miss some discussion regarding the motivation, why this research is performed and why it should matter to the readers.

The flow of the manuscript can also be improved, as sometimes I feel that the work is overloaded with formulas.

The standard bracket convention {[()]} is not followed, so formulas are hard to read as I have to count brackets.

Comments on the Quality of English Language

English is okay, though the grammar check finds some errors like missing/wrong articles or suggests rephrasing

Author Response

1. Introduction could be smoother, not all readers are skilled in the mathematics, so the first formula comming in the Introduction at the third line is a hard to swallow.
Yes, it is possible that the integral definition of the Airy function will put off some readers. But the very object of our study in this paper, being one of the simplest examples of non-Gaussian beams associated with diffraction catastrophe integrals, cannot be defined in a simpler way. However, taking into account the reviewer’s note, we have moved the specified formula from the first page to the second.
Generally speaking, to understand the properties of non-Gaussian beams during propagation (especially those related to diffraction catastrophe integrals), the use of hard mathematics cannot be avoided. This is the only way to predict the appearance of such effects during propagation as autofocusing, self-healing, bending trajectory, etc. Unfortunately for one of the authors (E.G.A.), we did not link the location of the autofocusing plane of three-Airy beams with a change in the growth order of this beam when propagating in the Fresnel zone (growth order is one of the most important characteristics of any entire analytical function; in this case, of two variables). Such a topic will certainly horrify the unprepared reader, although this approach works fine for next diffraction catastrophe beams (Pearcey, swallowtail, butterfly). Ignoring the theoretical description of an object studied in the natural sciences is like trying to win a fight by boxing with one hand. Why give the Nature such an advantage?
2. Next, I miss some discussion regarding the motivation, why this research is performed and why it should matter to the readers.
To be honest, the reasons why we wrote this article remained behind the scenes. We wanted to talk about the relationship between the growth order of the three-Airy beam and the location of its autofocusing plane. However, in the process of writing the article, we were able to solve this issue using simpler means. Therefore, the original structure of the article was redone. We have made the presentation more gentle for the reader. However, for multituple beams of Pearcey, Sw, Bu and higher, the use of growth order seems to be the only (at the moment) possibility to determine the locations of the autofocusing plane of non-Gaussian light fields, the intensity of which is far from circular.
Following the reviewer’s note, we have added the following paragraph at the end of the Introduction.
We consider the study of three-Airy beams at various values of the shift parameter, the characteristics of their propagation in the Fresnel zone, and the determination of the autofocusing plane depending on the shift parameter as a necessary basis for the subsequent transition to the study of multituple beams of Pearcey, swallowtail, and other diffraction catastrophes, whose non-Gaussian nature also leads to the appearance of an autofocusing plane when propagating in the Fresnel zone.
3. The flow of the manuscript can also be improved, as sometimes I feel that the work is overloaded with formulas.
Indeed, we could limit ourselves to considering the problem of finding the autofocusing plane, noting in the Conclusion that similar questions can be studied for other light beams connected with diffraction catastrophe integrals. However, in our view, more useful for the scientific community, young researchers and optics in general is a story about closely related problems, including the formulation and brief discussion of problems that have not yet been solved.
In our case, three-Airy beams are an nice example of this kind. And for further studies of generalized three-Airy beams, and as a basis for triple and muptiple beams of Pearcey, Sw, Bu, etc. In addition, we omit more difficult issues (growth order). However, we believe that the reader who is not afraid of mathematics and has studied the works of Berry, Nye, and Arnol’d will get a lot of useful information for his/her thinking and, perhaps, a desire to do something in the catastrophe optics. Actually, for this reason we presented the theoretical issues in sufficient detail.
4. The standard bracket convention [()] is not followed, so formulas are hard to read as I have to count brackets.
Following the reviewer’s note, we have changed brackets in Eqs.(43), (46), (A1), (A2), and (A3).
